# The *ACE* rs1799752 Variant Is Associated with COVID-19 Severity but Is Independent of Serum ACE Activity in Hospitalized and Recovered Patients

**DOI:** 10.3390/ijms24087678

**Published:** 2023-04-21

**Authors:** Ingrid Fricke-Galindo, Ivette Buendia-Roldan, Daniel I. Ponce-Aguilar, Gloria Pérez-Rubio, Leslie Chavez-Galan, Jesús Alanis-Ponce, Karina Pérez-Torres, Daniela Valencia-Pérez Rea, Fernanda Téllez-Quijada, Karol J. Nava-Quiroz, Rafael de Jesús Hernández-Zenteno, Angélica Gutiérrez-Nava, Ramcés Falfán-Valencia

**Affiliations:** 1HLA Laboratory, Instituto Nacional de Enfermedades Respiratorias Ismael Cosio Villegas, Mexico City 14080, Mexico; ingrid_fg@yahoo.com.mx (I.F.-G.); dannohoppers@gmail.com (D.I.P.-A.); glofos@yahoo.com.mx (G.P.-R.); 14alanisponce@gmail.com (J.A.-P.); daniela.980211@gmail.com (D.V.-P.R.); krolnava@hotmail.com (K.J.N.-Q.); 2Translational Research Laboratory on Aging and Pulmonary Fibrosis, Instituto Nacional de Enfermedades Respiratorias Ismael Cosío Villegas, Mexico City 14080, Mexico; ivettebu@yahoo.com.mx (I.B.-R.); mayode1997@hotmail.com (K.P.-T.); fertellezq@gmail.com (F.T.-Q.); 3Departamento de Sistemas Biológicos, División de Ciencias Biológicas y de la Salud, Universidad Autónoma Metropolitana Unidad Xochimilco, Mexico City 04960, Mexico; agutz@correo.xoc.uam.mx; 4Laboratory of Integrative Immunology, Instituto Nacional de Enfermedades Respiratorias Ismael Cosío Villegas, Mexico City 14080, Mexico; lchavez_galan@iner.gob.mx; 5COPD Clinic, Instituto Nacional de Enfermedades Respiratorias Ismael Cosío Villegas, Mexico City 14080, Mexico; rafherzen@yahoo.com.mx

**Keywords:** ACE, COVID-19, genetics, angiotensin-converting enzyme, single nucleotide variant, renin–angiotensin system

## Abstract

This paper assesses the association of the insertion/deletion *ACE* (angiotensin-converting enzyme) variant (rs1799752 I/D) and the serum ACE activity with the severity of COVID-19 as well as its impact on post-COVID-19, and we compare these associations with those for patients with non-COVID-19 respiratory disorders. We studied 1252 patients with COVID-19, 104 subjects recovered from COVID-19, and 74 patients hospitalized with a respiratory disease different from COVID-19. The rs1799752 *ACE* variant was assessed using TaqMan^®^ Assays. The serum ACE activity was determined using a colorimetric assay. The DD genotype was related to risk for invasive mechanical ventilation (IMV) requirement as an indicator of COVID-19 severity when compared to the frequencies of II + ID genotypes (*p* = 0.025, OR = 1.428, 95% CI = 1.046–1.949). In addition, this genotype was significantly higher in COVID-19 and post-COVID-19 groups than in the non-COVID-19 subjects. The serum ACE activity levels were lower in the COVID-19 group (22.30 U/L (13.84–32.23 U/L)), which was followed by the non-COVID-19 (27.94 U/L (20.32–53.36 U/L)) and post-COVID-19 subjects (50.00 U/L (42.16–62.25 U/L)). The DD genotype of the rs1799752 *ACE* variant was associated with the IMV requirement in patients with COVID-19, and low serum ACE activity levels could be related to patients with severe disease.

## 1. Introduction

The renin–angiotensin system (RAS) controls multiple functions in the cardiovascular system, including metabolism, cell growth, and homeostasis. RAS has been reported to participate in the inflammatory process of cardiac hypertrophy, pulmonary hypertension, glomerulonephritis, lung injury, sepsis, and acute pancreatitis [1].

The RAS is regulated by the angiotensin-converting enzyme (ACE) and its homologue angiotensin-converting enzyme 2 (ACE2) [2]. ACE is expressed in the lung, kidney, small intestine, brain, heart, adrenal gland, and gut. ACE interacts with angiotensin (Ang) I (Ang-I) to produce the vasoconstrictor Ang-II and inactivates bradykinin, producing vasoconstriction, fibrosis, inflammation, and thrombosis responses [3,4]. Meanwhile, ACE2 hydrolyzes Ang-II into Ang-1–7, which increases vasodilation [5]. In the lung, RAS regulates cell proliferation, inflammatory immune response, hypoxia, and angiogenesis [6].

In humans, *ACE* and *ACE2* genes are located at chromosomes 17q23 and Xp22. *ACE* is especially abundant in highly vascular organs such as the retina and lungs. Even the lung possesses the highest amount of ACE and contributes 0.1% of total protein [7]. *ACE* is under promoter regulation by hypoxia-inducing factor 1α (HIF-1α), which upregulates this gene expression under hypoxic conditions, increasing Ang-II levels. Under hypoxia, *ACE2* is downregulated through Ang-II [8].

The imbalance of this system leads to several complications observed in inflammatory syndromes [9]. Moreover, the regulation factor of *ACE* expression, HIF-1α, is affected by aging and high-risk factors (diabetes, hypertension, and chronic obstructive pulmonary disease) [10].

In coronavirus disease (COVID-19), the RAS seems to play an essential role in the pathogenesis of severe cases since the RAS is crucial to the homeostasis of both the cardiovascular and respiratory systems, and the severe acute respiratory syndrome coronavirus 2 (SARS-CoV-2) utilizes and interrupts this pathway directly [11,12]. SARS-CoV-2 infection is initiated by the specific binding of the viral spike protein to ACE2. The expression of *ACE2* and several genetic variants have been related to COVID-19 susceptibility and severity [13,14,15]. Likewise, high admission plasma ACE2 levels have been associated with increased COVID-19 severity within 28 days [16], and the serum ACE2 activity was related to COVID-19 severity and mortality risk in this disease [17].

Regarding ACE, different reports have linked *ACE* variants and their expression with COVID-19 mortality and severity among worldwide populations [18]. For instance, the rs1799752 *ACE* has been associated with COVID-19 risk and ACE expression [19,20] and with the ACE2 protein levels in alveolar lung epithelium in patients with the disease [21]. However, the ACE activity enzyme could be involved in the severity of COVID-19 and the outcome in post-COVID-19 patients, but this has not been previously evaluated. Thus, we aimed to assess the association of the insertion/deletion *ACE* variant (rs1799752) and the serum ACE activity with the severity of COVID-19 and its impact in post-COVID-19 subjects and patients hospitalized with a respiratory disease different from COVID-19.

## 2. Results

### 2.1. Clinical and Demographical Data of the COVID-19, Post-COVID-19, and Non-COVID-19 Subjects

The clinical and demographical characteristics of patients hospitalized with COVID-19 are shown in Table 1. Patients requiring invasive mechanical ventilation (IMV) were older and predominantly male (OR = 1.7, 95% CI = 1.3–2.2) compared to the non-IMV group. Co-morbidities such as type 2 diabetes mellitus (T2DM), systemic arterial hypertension (SAH), chronic respiratory diseases (CRD), and cardiovascular diseases (CVD) were observed in both groups and had a similar frequency. As expected, the PaO_2_/FiO_2_ levels and the length of hospital stay differed among groups, and the days since symptoms onset to hospital admission were slightly higher in the IMV group.

The post-COVID-19 group comprises patients discharged 5–12 months ago due to a COVID-19 diagnosis; they were predominantly males (65.0%), with a median age of 58 years, body mass index (BMI) of 28.2 kg/m^2^, and 32.5% of these patients still presented a respiratory dysfunction (see Section 4) when sampling was performed. Meanwhile, 55.4% of the non-COVID-19 group were males, and the median age was 63.5 years. Patients in this group were hospitalized for different respiratory, but non-infectious, diseases (i.e., cancer, vasculitis). Unfortunately, we could not obtain the complete clinical information for all non-COVID-19 patients, such as co-morbidities and disease severity.

### 2.2. Association of rs1799752 ACE with IMV Requirement in Patients with COVID-19

The allele and genotype frequencies of the rs1799752 variant are shown in Table 2. Neither the insertion (I) nor deletion (D) alleles were associated with the IMV requirement, while the frequency of the genotypes differed between the studied groups. The DD genotype was associated with the risk of requiring IMV in patients hospitalized with severe COVID-19. Individuals carrying the ID genotype could exhibit a low risk of needing the therapeutic procedure. These associations were also observed when the recessive and over-dominant models were performed (Table 2). These genetic associations remained in a logistic regression model adjusting for sex and age (Table 2), which are two covariates strongly related to the studied phenotype. We also evaluated if the genotypes were associated with mortality among a subgroup of patients in which data were available. Still, we did not find significant differences in the frequencies between survivor and non-survivor groups (Appendix A).

We also evaluated whether the genotype frequencies differed among the comparison groups (post-COVID-19 and non-COVID-19). The genotype distribution in the COVID-19 and post-COVID-19 groups did not accomplish the Hardy-Weinberg equilibrium (HWE); meanwhile, the frequencies observed in the non-COVID-19 group did meet the assumption. We found a higher frequency of DD genotype in the COVID-19 group when compared to non-COVID-19 (*p* = 0.031, OR = 2.67, 95% CI = 1.16–6.11) (Table 3), which could be related to a higher risk of severe COVID-19 in agreement with the previous result in which this genotype was associated with the IMV requirement, and the deviation from HWE found in COVID-19 groups; however, further studies in subjects with mild or moderate COVID-19 could drive to more robust conclusions. The DD genotype frequency also differed between post-COVID-19 and non-COVID-19 groups (*p* = 0.025, OR = 2.56, 95% CI = 0.99–6.62), but no differences were observed when comparing the frequencies in COVID-19 and post-COVID-19 (*p* = 0.431).

### 2.3. Serum ACE Activity Role in COVID-19 and Post-COVID-19

The serum ACE activity was assessed in available samples from patients of the three subgroups (COVID-19, post-COVID-19, and non-COVID-19). The accessible demographics and clinical information of these subgroups are included in Table 4.

The groups’ activity levels differed significantly (*p* < 0.001, Figure 1). We observed a higher activity level of enzyme activity in the post-COVID-19 group (50.00 U/L (42.16–62.25 U/L)) when compared to patients with pulmonary diseases (COVID-19 and non-COVID-19 groups). Meanwhile, lower levels were observed for the COVID-19 group (22.30 U/L (13.84–32.23 U/L)) than for the non-COVID-19 group (27.94 U/L (20.32–53.36 U/L)).

We wondered if post-COVID-19 patients still presenting respiratory anomalies were affecting the analysis; therefore, we performed the same analysis, including only those patients who completely recovered and reported as healthy when the sampling was performed. The significant values present minimal variations that can be observed in Appendix A.

Differences in serum ACE activity levels were independent of the rs1799752 *ACE* genotypes. Figure 2 shows that the enzyme activity levels were not statistically different according to genotypes among each clinical group (COVID-19 *p* = 0.165, post-COVID19 *p* = 0.288, non-COVID-19 *p* = 0.226, Kruskal–Wallis test). Thus, we wondered if there were some risk factors impacting the serum ACE activity and if this was related to the requirement of IMV. Among the COVID-19 group, sex, age, BMI, co-morbidities, and the rs1799752 genotype were unrelated to the activity levels (Appendix A). Nevertheless, a tendency toward difference (*p* = 0.057) was observed when the activity levels were compared between the patients requiring IMV (*n* = 43, 19.12 U/L (11.76–29.29 U/L)) and non-IMV (*n* = 23, 28.43 U/L (17.65–41.91 U/L)), suggesting that low serum ACE activity is related with higher COVID-19 severity. However, we did not observe a trend with other clinical parameters such as IMV days or PaO_2_/FiO_2_ (Appendix A). We also performed a logistic regression model using IMV as a dependent variable, and the variables included were serum ACE activity, sex, and age, and the only variant that was found significant was sex (*p* = 0.019); therefore, further studies are required to determine if there are additional factors that could be affecting the apparent difference in the ACE activity among IMV and non-IMV groups.

We also evaluated if the serum ACE activity level was related to available demographic characteristics and/or clinical parameters in the post-COVID-19 and non-COVID-19 groups. In the post-COVID-19 group, the ACE activity level was not different according to sex (*p* = 0.322), age (*p* = 0.841), BMI (*p* = 0.144), months since patients’ discharge (*p* = 0.942), co-morbidities (T2DM *p* = 0.609; SAH *p* = 0.573; heart diseases *p* = 0.858), nor if patients still presented a respiratory dysfunction (*p* = 0.407). In patients with non-COVID-19, we found a correlation between the ACE activity with age (*p* = 0.042, rho = −0.829), but no differences in the activity levels were observed according to sex (*p* = 0.602). A multivariate analysis was not performed due to the lack of associations in univariate models.

## 3. Discussion

The COVID-19 pandemic has uncovered several questions about the inter-individual variability in the severity and clinical outcomes of the disease. Different risk factors have been related to the complications of subjects with COVID-19, including genetic and non-genetic issues. Herein, we report the association of the DD genotype of rs1799752 *ACE* with the IMV requirement among patients with severe COVID-19 and that the serum ACE activity is decreased in patients coursing with COVID-19 when compared to subjects with other pulmonary diseases and convalescent COVID-19 patients.

The association of the *ACE* rs1799752 variant with COVID-19 severity has been observed in different reports. A recent meta-analysis, including 11 studies with 950 patients with severe COVID-19, and 1573 patients with non-severe disease, reported that the DD genotype was related to COVID-19 severity through different models [22]. This finding agrees with our study and other reports in diverse populations [19,20,23] as well as another meta-analysis [24], which highlights the relevance of the RAS system, mainly the *ACE* gene, in the severity of COVID-19 and other pulmonary diseases. According to our findings, the DD genotype confers a risk of IMV requirement; meanwhile, the heterozygous genotype (ID) shows a decreased risk of requiring the procedure. The risk finding of the DD genotype is in agreement with the previous reports, and the presence of both D alleles is required to present the risk, as it was observed in the additive and over-dominant models, where the presence of the insertion in the ID genotype shows a decreased risk of requiring IMV among patients with COVID-19.

The differences in the DD frequencies among COVID-19/post-COVID-19 and non-COVID-19 groups could be controversial due to the sample size, or the DD genotype could provide susceptibility to COVID-19, as it was associated with susceptibility of COVID-19 in a meta-analysis including 13 studies [25]. Unfortunately, we could not determine the frequency in a group of healthy volunteers of Mexican Mestizo origin. Nevertheless, we evaluated if our frequencies differed from those reported in a pre-pandemic study by Vargas-Alarcon et al., 2003 [26], including 98 Mexican Mestizos. We did not find any differences with our studied groups (vs. COVID-19 group *p* = 0.763, vs. post-COVID-19 *p* = 0.391, and vs. non-COVID-19 *p* = 0.200). Thus, an indirect association with the risk of severe COVID-19 could be observed, which aligns with the HWE deviation found in the COVID-19 groups but not in the non-COVID-19 group.

The deletion allele has also been related to an increased risk of hypertension, pre-eclampsia, heart failure, cerebral infarct, diabetic nephropathy, encephalopathy, asthma, severe hypoglycemia in diabetes, gastric cancer (in Caucasians) and poor prognosis following kidney transplant [27]. In agreement, we found an association with complicated conditions such as the IMV requirement in patients with severe COVID-19. The D allele has been linked with increased activity of the ACE enzyme [27], which we observed in the post-COVID-19 group (Figure 2) but not in the COVID-19 group. We thought that risk factors related to COVID-19 severity (i.e., age, sex, and co-morbidities) acted as confounders; however, the statistical analyses showed that the serum ACE activity was not related to any of these variables. In this sense, only in the non-COVID-19 group, we observed a strong negative correlation between enzyme activity and age, but this could not be applied to the other groups. However, this finding could indicate that ACE is involved and affected during SARS-CoV-2 infection independently of other factors.

Previous studies have reported the relevance of ACE activity in the severity of COVID-19 [28]. Guler et al., reported similar serum ACE activity between 55 patients with COVID-19, including asymptomatic, mild, and severe groups, and 18 controls [29]. Meanwhile, the study of Reindl-Schwaighofer et al., reported a significantly lower plasma ACE activity, measured as angiotensin II/I ratio, related to severe COVID-19, and that did not correlate with ACE concentrations in plasma [30]. In this sense, lower serum ACE activity can be observed in patients with severe COVID-19, and activity levels are restored when patients are recovered (post-COVID-19 group). This finding could be supported by the report of higher ACE concentration in children and adolescents than adults [31], which are two populations with a low risk of presenting COVID-19 complications. On the other hand, an increment in serum ACE activity may appear as a response to chronic hypoxia due to respiratory dysfunction, as it has been observed in patients with emphysema, extrinsic asthma, and small cell carcinoma of the lung [32]; however, a purposeful study including more patients is warranted.

Moreover, we found that the ACE activity level is impaired in patients with other respiratory diseases but not in the same proportion as the COVID-19 group, since the non-COVID-19 group exhibited a lower enzyme activity when compared to the post-COVID-19 group. However, differential enzyme activity levels have been reported according to respiratory diseases. For instance, higher ACE activity levels have been reported in patients with sarcoidosis but lower levels have been reported in those with fibrosing alveolitis, interstitial lung disease, and chronic obstructive lung disease [33]. Unfortunately, we could not obtain the complete clinical information of patients in the non-COVID-19 group to draw more solid conclusions.

This study highlights the relevance of *ACE* and serum ACE activity in COVID-19 and other respiratory diseases. Nevertheless, this study is not exempt from limitations. First, evaluating serum ACE activity with non-infected and healthy subjects would be interesting as well as considering the influence of pharmacological treatment on enzyme activity. We also acknowledge the small sample size of patients in whom the serum ACE activity was determined, which could require further confirmation studies. In the non-COVID-19 group, ample clinical information could drive additional information about the relevance of serum ACE activity in non-infectious respiratory diseases. Finally, it is worth mentioning that ACE is significantly expressed in the lung. This information could provide valuable insight into current and future infectious illnesses and other chronic respiratory diseases.

## 4. Materials and Methods

### 4.1. Subjects

This study was performed in the Instituto Nacional de Enfermedades Respiratorias Ismael Cosio Villegas (INER) (Mexico City, Mexico). For the study design, three groups of subjects were included: (1) subjects with a diagnosis of COVID-19 confirmed by real-time polymerase chain reaction test and hospitalized in the institute (*n* = 1252, recruited from July 2020 to February 2021); (2) subjects followed after their hospitalization due to COVID-19 (post-COVID-19, *n* = 104, recruited from March to October 2021); and (3) patients hospitalized with a respiratory disease but non-COVID-19 diagnosis (*n* = 74, recruited from January to March 2022). All patients were ≥ 18 years old, and they, or a responsible family member, signed the informed consent. The study protocol was approved by the local Research Ethics Committee (C53-20) and complied with the Helsinki Declaration criteria.

The severe disease in patients with COVID-19 was determined by the presence of dyspnea, creating a respiratory rate of ≥30 breaths per minute, blood oxygen saturation ≤ 90%, and/or PaO_2_/FiO_2_ ≤ 300 at the hospital admission [34]. The post-COVID-19 group includes patients who were followed up after their hospital discharge since they presented a pulmonary dysfunction determined by a decrease in their forced vital capacity or desaturation in the 6 min walking test and/or the presence of interstitial thickening in the computed tomography, which has been previously reported in other cohorts [35,36]. All patients in this group coursed severe COVID-19 when hospitalized and required respiratory support through IMV or high-flow nasal cannula oxygen therapy. Fifty-nine subjects of the post-COVID-19 group were included in the COVID-19 group; meanwhile, 45 patients were hospitalized in the INER due to severe COVID-19 but were not in the COVID-19 group.

Blood sampling was performed during the hospital stay of the subjects with COVID-19 and non-COVID-19. Subjects with post-COVID-19 were sampled during one of their follow-up medical consultations. Demographic and clinical data were acquired from medical records and the clinical interview with outpatients. We considered the IMV requirement (IMV and non-IMV) as a severity indicator and dependent variable for the association study.

### 4.2. Genotyping

DNA was isolated from peripheral blood collected in tubes with EDTA, employing the commercial BDtract Genomic DNA isolation kit (Maxim Biotech, San Francisco, CA, USA), and stored at 4 °C until processing. The *ACE* rs1799752 was determined through the TaqMan^®^ SNP Genotyping Assays C_60538594A_10 and C_60538594B_20; the former reports the mutant Alu insertion allele (FAM probe), and the latter reports the wild-type deletion allele (VIC probe) of the *ACE* I/D variant. The analyses were performed according to the supplier instructions in a 7300 Real-Time PCR System (Applied Biosystems/ThermoFisher Scientific Inc., Marsiling, Singapore).

### 4.3. Serum ACE Activity

The serum ACE activity was determined in a subset of patients from each group (COVID-19 *n* = 66, post-COVID-19 *n* = 69, and non-COVID-19 *n* = 26). Patients’ samples from COVID-19 and post-COVID-19 groups were selected according to (a) serum sample availability and (b) the rs1799752 *ACE* genotype to ensure that samples from patients with the three genotypes were assessed for ACE activity. In the non-COVID-19 group, all the serum samples available were included independently of the genotype. Serum was separated from peripheral blood collected in tubes without anticoagulant by centrifugation and stored at −80 °C until use. The ACE activity was assessed employing the ACE Activity Assay Kit (Colorimetric) ab273308 (Abcam, Cambridge, UK) according to the supplier recommendations. Reads were performed at two wavelengths (345 and 600 nm) every 10 min for one hour. The enzyme activity was determined with the reads from 10 and 60 min, following the equation indicated by the supplier, the ACE activity, in U/L. Only samples and/or positive control with a correlation coefficient > 0.8 were considered. Blank, positive control, and samples were evaluated in duplicate, and the mean of each well was reported.

### 4.4. Statistical Analysis

According to the data distribution, categorical data are reported as frequencies, and continuous values are presented as mean ± standard deviation or median (interquartile range). A Kolmogorov–Smirnov test was employed for the normality assessment. HWE was assessed using a chi-square test. The association of *ACE* rs1799752 with the IMV requirement was assessed using PLINK v 1.07 [37]. The comparison of categorical variables was performed with the chi-square test or Fisher’s exact test; meanwhile, the evaluation of continuous data was performed with Mann–Whitney U or Kruskal–Wallis test with Benjamini–Hochberg correction, as required. A logistic regression model was employed to adjust for covariates in the association analyses. Statistics were analyzed using RStudio Workbench 2022.07.2 [38], the ggplot2 [39], ggpubr, and ggsignif packages.

## Figures and Tables

**Figure 1 ijms-24-07678-f001:**
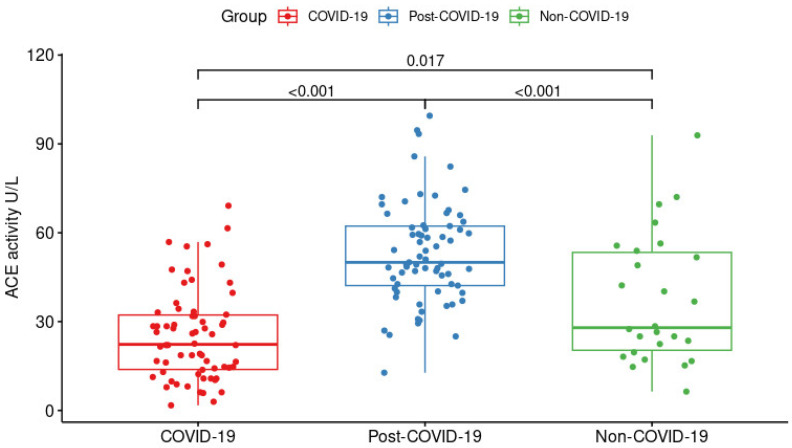
Serum ACE activity levels in the COVID-19 (*n* = 66), post-COVID-19 (*n* = 69), and non-COVID-19 (*n* = 26) groups. The comparisons were performed using the Kruskal–Wallis test corrected with the Benjamini–Hochberg method.

**Figure 2 ijms-24-07678-f002:**
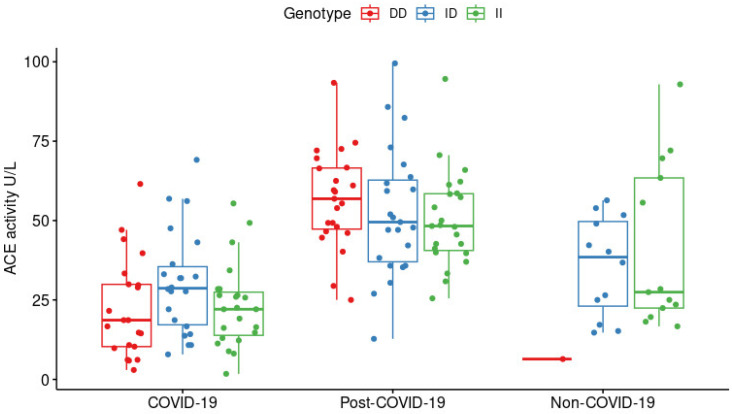
Serum ACE activity according to the rs1799752 *ACE* genotypes in the COVID-19 (*n* = 66), post-COVID-19 (*n* = 69), and non-COVID-19 (*n* = 26) groups. The ACE activity was not different among the three genotypes in any of the studied groups (Kruskal–Wallis test, *p* > 0.05).

**Table 1 ijms-24-07678-t001:** Clinical and demographic data of patients with severe COVID-19.

Variable ^a^	All (*n* = 1252)	IMV (*n* = 899)	Non-IMV (*n* = 353)	*p*-Value ^b^
Age, years	59 (50–67)	60 (51–68)	56 (48–65)	<0.001
Sex, M/F (%)	844/408 (67.4/32.6)	638/261 (71.0/29.0)	206/147 (58.4/41.6)	<0.001
BMI, kg/m2	29.3 (26.2–33.2)	29.4 (26.6–33.3)	28.9 (25.7–33.1)	0.103
Smokers (%)	363 (29.0)	264 (29.4)	99 (28.1)	0.564
Co-morbidities				
T2DM (%)	340 (27.2)	246 (27.4)	94 (26.8)	0.561
SAH (%)	438 (35.0)	318 (35.4)	120 (34.0)	0.817
CRD (%)	95 (7.6)	68 (7.6)	27 (7.6)	0.234
CVD (%)	49 (3.9)	38 (4.3)	11 (3.1)	0.178
PaO2/FiO2	147 (102.8–196.2)	135 (94.2–178)	200 (141–240)	<0.001
Days since symptoms onset	9 (7–13)	10 (7–14)	9 (7–12)	0.035
Hospital stay, days	18 (12–29)	22 (16–33)	11 (8–15)	<0.001
IMV days	NA	17.5 (11–28)	NA	NA
Non-survivors	381 (33.9)	354 (44.5)	27 (8.2)	<0.001

Data are presented as median (interquartile range) and absolute account (percentage) for continuous and categorical variables. ^a^ Clinical data were unavailable for some subjects; ^b^ The comparisons were performed using the Mann–Whitney U and Fisher’s exact tests. BMI, body mass index; CRD, chronic respiratory diseases; CVD, cardiovascular diseases; M, male; F, female; IMV, invasive mechanical ventilation; SAH, systemic arterial hypertension; T2DM, type 2 diabetes mellitus.

**Table 2 ijms-24-07678-t002:** Association study of the rs1799752 *ACE* with the requirement of IMV among patients with COVID-19.

Allele/Genotype	IMV (*n* = 899)	Non-IMV (*n* = 353)	*p*-Value ^a^	OR (95% CI)	Adjusted *p*-Value ^b^
I	1045 (0.581)	422 (0.598)	0.450		
D	753 (0.419)	284 (0.402)		NA	NA
II	362 (0.403)	133 (0.377)		1	
ID	321 (0.357)	156 (0.442)	0.010	0.756 (0.574–0.996)	0.011
DD	216 (0.240)	64 (0.181)		1.240 (0.880–1.746)	
Recessive model					
II + ID	683 (0.760)	289 (0.819)	0.025		
DD	216 (0.240)	64 (0.181)		1.428 (1.046–1.949)	0.027
Dominant model					
II	362 (0.403)	133 (0.377)			
ID + DD	537 (0.597)	220 (0.623)	0.399	NA	NA
Over-dominant model					
ID	321 (0.357)	156 (0.442)	0.005		
II + DD	578 (0.643)	197 (0.558)		0.701 (0.546–0.901)	0.006

Frequencies are presented as absolute counts and percentages (proportion). ^a^ Chi-square test; ^b^ logistic regression model using age and sex as covariates. 95% CI, 95% confidence interval; D, deletion; I, insertion; IMV, invasive mechanical ventilation; OR, odds ratio.

**Table 3 ijms-24-07678-t003:** Comparison of the genotype frequencies of the rs1799752 *ACE* among the studied groups.

Genotype	COVID-19 (*n* = 1252) ^a^	Post-COVID-19 (*n* = 104) ^a^	Non-COVID-19 (*n* = 74) ^c^
II	495 (0.395)	46 (0.442)	33 (0.446)
ID	477 (0.381)	33 (0.317)	34 (0.459)
DD	280 (0.224) ^b^	25 (0.240) ^b^	7 (0.095) ^b^

Data are presented as n (frequency). D, deletion; I, insertion. ^a^ The COVID-19 and post-COVID-19 groups presented a deviation from Hardy–Weinberg equilibrium; ^b^ The DD genotype frequencies were different when compared COVID-19 vs. non-COVID-19 (*p* = 0.031) and post-COVID-19 vs. non-COVID-19 (*p* = 0.025); ^c^ The frequencies in the non-COVID-19 group were according to the Hardy–Weinberg equilibrium.

**Table 4 ijms-24-07678-t004:** Demographic and clinical characteristics of patients included in the subgroups for the serum ACE activity determination.

Variable	COVID-19 (*n* = 66)	Post-COVID-19 (*n* = 69)	Non-COVID-19 (*n* = 26)	*p*-Value ^a^
Age, years	61.5 (52.2–69.8)	58 (50–65.2)	69.5 (37–76)	0.224
Sex, M/F (%)	45/21 (68.2/31.8)	49/20 (71.0/29.0)	13/13 (50.0/50.0)	0.086
BMI, kg/m^2^	27.9 (25.7–31.2)	28 (26.1–31.8)	NA	0.666
Co-morbidities				
T2DM (%)	22 (33.3)	22 (31.9)	NA	0.857
SAH (%)	22 (33.3)	18 (26.1)	NA	0.357
CRD (%)	4 (6.1)	6 (8.7)	NA	0.559
CVD (%)	3 (4.5)	2 (2.9)	NA	0.612

^a^ Kruskal–Wallis test or chi-square test. Continuous data are presented as median (interquartile range). BMI, body mass index; CRD, chronic respiratory diseases; CVD, cardiovascular diseases; F, female; IMV, invasive mechanical ventilation; M, male; NA, not available data; SAH, systemic arterial hypertension; T2DM, type 2 diabetes mellitus.

## Data Availability

Data available at SCV002818477.

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
