# Peer review of "The ACE rs1799752 Variant Is Associated with COVID-19 Severity but Is Independent of Serum ACE Activity in Hospitalized and Recovered Patients"

_ijms, 2023, doi:10.3390/ijms24087678_

Round 1
Reviewer 1 Report
In this article, the authors deal with the correlation between ACE variant rs1799752 I/D and the serum ACE activity with COVID-19 severity. The manuscript is complex and well written, but there are still some points to be addressed by the authors in order to improve their manuscript.
General comments
Introduction is unnecessarily long, and should be restricted to the descriptions related to the present study.
The major concern is that the number of samples used in this study to determine the serum ACE activity role was too small to draw a plausible conclusion.
Author Response
In this article, the authors deal with the correlation between ACE variant rs1799752 I/D and the serum ACE activity with COVID-19 severity. The manuscript is complex and well written, but there are still some points to be addressed by the authors in order to improve their manuscript.
Authors’ response: We thank you for the valuable comments of the Reviewer.
General comments
Introduction is unnecessarily long, and should be restricted to the descriptions related to the present study.
Authors’ response: Attending your suggestion, we have summarized the Introduction section as possible.
The major concern is that the number of samples used in this study to determine the serum ACE activity role was too small to draw a plausible conclusion.
Authors’ response: We agree with the Reviewer’s comment. This was remarked as a limitation of the study (lines 264-265) and in the Abstract conclusion.
Reviewer 2 Report
This study evaluates the ACE (I/D) gene variant and ACE activity in patients with coronavirus disease (COVID-19) and post-COVID-19 patients and patients with non-COVID-19 related respiratory disorders.
The authors found that the DD genotype is related to the risk of requiring invasive mechanical ventilation (IMV) as an indicator of COVID-19 severity compared to the frequencies of the II + ID genotypes. The frequency of the DD genotype was significantly higher in the COVID-19 and post-COVID-19 groups than in the non-COVID-19 subjects. The authors found significant differences in the age and sex distributions, but the genetic associations remained in their logistic regression model adjusting for these variables.
The main concern is that the differences in DD frequencies between the COVID-19/Post-COVID-19 and non-COVID-19 groups could be due to sample size. However, the authors adequately discuss this limitation. Surprisingly, they found higher ACE enzyme activity associated with the presence of the D allele in the post-COVID-19 group, but not in the COVID-19 group. However, these observations are discussed in accordance with recent findings and, as cited, decreased serum ACE activity has been observed in patients with severe COVID-19 but activity levels are restored when patients recover from the disease. In addition, an important limitation is the absence of a control group. Although the arguments that justify it are somewhat weak, the authors, warning of the need for confirmatory studies, also point out this limitation.
Author Response
REVIEWER 2
This study evaluates the ACE (I/D) gene variant and ACE activity in patients with coronavirus disease (COVID-19) and post-COVID-19 patients and patients with non-COVID-19 related respiratory disorders.
The authors found that the DD genotype is related to the risk of requiring invasive mechanical ventilation (IMV) as an indicator of COVID-19 severity compared to the frequencies of the II + ID genotypes. The frequency of the DD genotype was significantly higher in the COVID-19 and post-COVID-19 groups than in the non-COVID-19 subjects. The authors found significant differences in the age and sex distributions, but the genetic associations remained in their logistic regression model adjusting for these variables.
The main concern is that the differences in DD frequencies between the COVID-19/Post-COVID-19 and non-COVID-19 groups could be due to sample size. However, the authors adequately discuss this limitation. Surprisingly, they found higher ACE enzyme activity associated with the presence of the D allele in the post-COVID-19 group, but not in the COVID-19 group. However, these observations are discussed in accordance with recent findings and, as cited, decreased serum ACE activity has been observed in patients with severe COVID-19 but activity levels are restored when patients recover from the disease. In addition, an important limitation is the absence of a control group. Although the arguments that justify it are somewhat weak, the authors, warning of the need for confirmatory studies, also point out this limitation.
Authors’ response: We thank you for the valuable and positive comments of the Reviewer. We have highlighted the importance of confirmatory studies to draw more robust conclusions.
Reviewer 3 Report
Dear Editor,
I have revised the manuscript entitled “The ACE rs1799752 variant is associated with COVID-19 severity but is independent of serum ACE activity in hospitalized and recovered patients” by Ingrid Fricke-Galindo et al. The topic is important, the study is properly designed and the results are well presented.
Minor issues
- lines 32-33 “The DD genotype of rs1799752.” needs a verb
- references – the format is not consistently used.
Sincerely yours,
Author Response
I have revised the manuscript entitled “The ACE rs1799752 variant is associated with COVID-19 severity but is independent of serum ACE activity in hospitalized and recovered patients” by Ingrid Fricke-Galindo et al. The topic is important, the study is properly designed and the results are well presented.
Authors’ response: We want to thank the Reviewer for the positive comments and the suggestions. We appreciate the time and the valuable review performed.
Minor issues
- lines 32-33 “The DD genotype of rs1799752.” needs a verb.
Authors’ response: We thank the Reviewer for the observation. The point interrupting the sentence was deleted. We apologize for the mistake.
- references – the format is not consistently used.
Authors’ response: The references were verified and corrected according to the journal reference style.